# Elevation of Arginase-II in Podocytes Contributes to Age-Associated Albuminuria in Male Mice

**DOI:** 10.3390/ijms241311228

**Published:** 2023-07-07

**Authors:** Guillaume Ajalbert, Andrea Brenna, Xiu-Fen Ming, Zhihong Yang, Duilio M. Potenza

**Affiliations:** Laboratory of Cardiovascular and Aging Research, Department of Endocrinology, Metabolism and Cardiovascular System, Faculty of Science and Medicine, University of Fribourg, Chemin du Musée 5, CH-1700 Fribourg, Switzerland; guillaume.ajalbert@unifr.ch (G.A.); andrea.brenna@unifr.ch (A.B.); xiu-fen.ming@unifr.ch (X.-F.M.)

**Keywords:** albuminuria, arginase, kidney aging, podocytes

## Abstract

One of the manifestations of renal aging is podocyte dysfunction and loss, which are associated with proteinuria and glomerulosclerosis. Studies show a male bias in glomerular dysfunction and chronic kidney diseases, and the underlying mechanisms remain obscure. Recent studies demonstrate the role of an age-associated increase in arginase-II (Arg-II) in proximal tubules of both male and female mice. However, it is unclear whether Arg-II is also involved in aging glomeruli. The current study investigates the role of the sex-specific elevation of Arg-II in podocytes in age-associated increased albuminuria. Young (3–4 months) and old (20–22 months) male and female mice of *wt* and arginase-II knockout (*arg-ii^−/−^*) were used. Albuminuria was employed as a readout of glomerular function. Cellular localization and expression of Arg-II in glomeruli were analyzed using an immunofluorescence confocal microscope. A more pronounced age-associated increase in albuminuria was found in male than in female mice. An age-associated induction of Arg-II in glomeruli and podocytes (as demonstrated by co-localization of Arg-II with the podocyte marker synaptopodin) was also observed in males but not in females. Ablation of the *arg-ii* gene in mice significantly reduces age-associated albuminuria in males. Also, age-associated decreases in podocyte density and glomerulus hypertrophy are significantly prevented in male *arg-ii^−/−^* but not in female mice. However, age-associated glomerulosclerosis is not affected by *arg-ii* ablation in both sexes. These results demonstrate a role of Arg-II in sex-specific podocyte injury in aging. They may explain the sex-specific differences in the development of renal disease in humans during aging.

## 1. Introduction

Age is the predominant risk factor for renal diseases. With aging, the kidney becomes more vulnerable to acute and chronic insults such as ischemia, dehydration, nephrotoxic drugs and toxins, etc., resulting in renal damage and failure [1,2]. Aging kidney is characterized by nephrosclerosis, a combination of two or more histologic features, including any global glomerulosclerosis, tubular atrophy, interstitial fibrosis > 5%, and arteriosclerosis, podocyte injury or a decrease in podocyte number, resulting in the dysfunctional integrity of the glomerular filtration barrier [3]. Approximately 73% of healthy kidney donors over 70 show nephrosclerosis [4]. In accordance with this, renal functions decline with age, which is manifested as a decline in glomerular filtration rate (GFR), proteinuria, and decreased tubular function such as reduced sodium reabsorption, potassium excretion, and urine-concentrating capacity [3]. Clinical studies show a male bias for glomerular dysfunction and a faster progression of chronic kidney diseases in men than in women with aging [5,6], hinting at differences in glomerular aging mechanisms. Exploring the molecular mechanisms of renal aging could shed light on novel therapeutic targets for age-associated renal diseases.

Arginase II (Arg-II), the extrahepatic and mitochondrial isoform of ureohydrolase that metabolizes L-arginine to urea and L-ornithine [7,8], is highly expressed in the kidney and pancreas [9]. In the kidney, Arg-II is predominantly expressed in S3 proximal tubular epithelial cells [10], but it has also been detected in other renal cells, including collecting duct principal cells [11] and glomerular cells, such as podocytes [12] and glomerular endothelial cells [13]. While the physiological role of Arg-II in the kidney remains largely unknown, an elevated level of Arg-II in the kidney has been shown to contribute to various renal pathologies, including aging [13,14,15,16,17].

Elevated Arg-II has been shown to impact the aging process of many organs, including the kidney [16], vasculature [18], heart [19], pancreas [20], and lung [21]. Particularly in the kidneys, the age-related elevation of Arg-II in the proximal renal tubules plays a major role in aging processes in a natural mouse aging model [16]. In addition, aging phenotypes of many organs (including kidneys) are different between males and females, which has been associated with the sex-specific expression of Arg-II levels in cells/organs [16,20,21]. Moreover, it was recently shown that Arg-II in podocytes mediates hypoxia-induced podocyte injury [12]. Interestingly, in line with human studies [5,6] which show a male bias for glomerular dysfunction and the faster progression of chronic kidney diseases with aging, more pronounced podocyte dysfunction in male horses and cotton rats has been reported [22,23]. However, it remains unknown as to the mechanisms underlying this sex difference. Given the essential role of podocytes in glomerular filtration barrier integrity [24], and considering that Arg-II may potentially be upregulated in injured podocytes [23], this work aimed to investigate whether Arg-II is elevated with aging in podocytes, contributing to the age-associated increase in albuminuria in a natural aging mouse model.

## 2. Results

### 2.1. arg-ii Ablation Decreases Age-Associated Albuminuria in Male but Not Female Mice

Urinary albumin but not creatinine and, therefore, the albumin/creatinine (uACR) ratio were significantly enhanced in both male and female *wt* old (20–22 months) mice when compared to the respective young groups (3–5 months; Figure 1).

The age-related increases in urinary albumin and uACR were overall more pronounced in the males when compared to the females (Figure 1B,C vs. Figure 1E,F). *arg-ii* ablation significantly reduced the age-associated increase in albuminuria in male mice (Figure 1B,C). In contrast, the age-associated increase in albuminuria in age-matched female animals was not affected by *arg-ii* gene deficiency (Figure 1E,F). Of note, plasma creatinine concentrations were comparable among young and old mice of both sexes in *wt* and *arg-ii^−/−^* animals (Figure 2).

### 2.2. arg-ii Ablation Protects Male Mice from Age-Associated Decreases in Podocytes

The number of podocytes, i.e., Wilms tumor protein 1-positive (WT1^+^) cells, in the glomeruli was decreased in *wt* old male mice as compared to the young male animals (Figure 3A,B). *arg-ii^−/−^* ablation did not affect the podocyte number in the young mice but significantly prevented the decrease in podocyte number in the aged mice of the male group (Figure 3A,B). Moreover, the age-associated glomerular hypertrophy as measured by glomerular area was significantly reduced in *arg-ii^−/−^* male mice (Figure 3C). In contrast, the age-associated decrease in podocyte number and the age-associated glomerular hypertrophy were not affected by *arg-ii^−/−^* in female mice (Figure 3D–F). This observation suggests that *arg-ii* may, at least partially, mediate age-related podocyte loss in males but not females.

### 2.3. arg-ii Ablation Has No Effects on Age-Associated Glomerulosclerosis in Both Sexes

Periodic acid–Schiff (PAS) staining demonstrated an age-associated increase in glomerulosclerosis at different levels of severity as analyzed by scores of 1 to 4. As reported in Figure 4, a score between 0 and 1 was assigned to the glomeruli of young groups (>70%). In old groups of both sexes, there were increases in the prevalence of scores above 2. However, this was not influenced by *arg-ii* ablation, suggesting that *arg-ii* did not affect age-related glomerulosclerosis in males or females.

### 2.4. Age-Dependent Increase in Arg-II Level in Podocytes in Male but Not in Female Mice

Kidney sections of both male and female mice (young and old groups) were immunostained using a specific anti-Arg-II antibody to investigate Arg-II protein levels in glomeruli. Figure 5A shows that Arg-II is not present in the glomeruli of young mice (neither male nor females) and that old *wt* male glomeruli display an age-mediated upregulation of this enzyme (green color; Figure 5A,B). Furthermore, co-immunostaining of Arg-II and the podocyte marker synaptopodin (SNPT) reveals that Arg-II protein partially co-localizes in the podocytes of old *wt* kidneys, as demonstrated by the physical overlapping of Arg-II (green) and SNPT (red) fluorescent signals (see the enlarged image in Figure 5C). Together, these results suggest that Arg-II is induced by aging in podocytes, as well as in other glomerular cells in male but not female mice.

## 3. Discussion

The current study demonstrates a sex-specific elevation of Arg-II in podocytes during aging, contributing to the decline in podocyte density and glomerular dysfunction as reflected by albuminuria in male mouse.

The kidney comprises discrete cell types, including tubular epithelial cells, glomerular endothelial cells, smooth muscle cells, podocytes, pericytes, mesangial cells, and macula densa cells [24]. In the kidney, Arg-II is predominantly expressed in S3 proximal tubular epithelial cells [10], but is inducible in other renal cell types, including collecting duct principal cells [11] and glomerular cells, such as podocytes [12] and glomerular endothelial cells [13]. Studies indicate that Arg-II in different renal cell types may play specific roles in renal physiology and/or pathophysiology. A previous study showed that Arg-II in collecting duct principal cells under water deprivation negatively regulates renal aquaporin-2 and water reabsorption [11]. Pathologically, the ablation of *arg-ii* in the whole body protects mice from renal ischemic/reperfusion injury [25]. It has also been reported that conditional ablation of Arg-II in endothelial cells reduces renal fibrosis in a unilateral ureteral obstruction mouse model [13]. In podocytes, Arg-II is not detectable under physiological conditions, but its expression is induced by hypoxia and plays an important role in type 2 diabetic nephropathy [12,26] and hypoxia-induced podocyte injury through mitochondrial dysfunction [12]. However, the role of Arg-II in aging-associated podocyte injury and proteinuria was not explored. Here, this study shows that Arg-II levels in a naturally aging mouse are elevated in podocytes along with decreased podocyte density, glomerular hypertrophy, and increased albuminuria in old male mice, which is significantly prohibited in age-matched old *arg-ii^−/−^* animals. In contrast with male mice, this aging phenotype is not affected by *arg-ii^−/−^* in females, suggesting a critical role of Arg-II in age-associated podocyte dysfunction, leading to enhanced proteinuria in male but not female animals. In support of this conclusion, Arg-II is induced and elevated in glomerular cells, including podocytes, in males but not in females. While the age-associated increase in total Arg-II level in the kidney was observed in both males and females [16], the age-associated elevation of Arg-II in podocytes occurs only in male mice. In accordance with this, the protective effects of *arg-ii^−/−^* on the age-related decrease in podocyte density, glomerular hypertrophy, and increase in proteinuria are observed in male mice but not in female mice. These results suggest a role of Arg-II in age-associated podocyte dysfunction predominantly in males.

It is widely recognized that focal segmental glomerulosclerosis (FSGS), a type of kidney disease characterized by scarring of the glomeruli and a leading cause of excess protein loss, is associated with glomerular podocyte damage [27]. However, a protective effect of Arg-II-ablation on age-associated glomerulosclerosis was not observed in both sexes in the current study. One possible explanation is that age-associated glomerulosclerosis involves multiple (including Arg-II-independent) mechanisms.

Sex-specific effects of genes have been documented, particularly regarding lifespan. It is widely recognized that human females live an average of 4–10 years longer than males [28]. Moreover, the sex-specific effects of genetic interventions on longevity have been reported very often in mice. For example, the genetic ablation of IGF1R [29], IRS1 [30], S6K1 [31], or Arg-II [19] enhances longevity only or prominently in female mice. In contrast, the effect of the genetic intervention of Rictor [32] or Sirt6 [33] on longevity is only observed in male mice. Specifically, several observations indicate a sex bias of Arg-II expression levels under physiological and/or pathological conditions and its detrimental effects in various tissues. Higher Arg-II levels have been observed in the kidney [16], heart [19], skin [19], pancreas [20], and lung [21] of females when compared to male mice. In contrast, the current study demonstrated that an age-associated increase in Arg-II in podocytes is only detected in male mice. The observed pathological phenotypes are associated with Arg-II levels in various organs, linking elevated Arg-II to related pathologies [16,19,20,21]. A possible explanation for this sex-specific phenomenon may lie in differential hormonal patterns between male and female animals. Indeed, podocytes are known to be target cells for testosterone and 17β-estradiol, with the former inducing apoptosis and the latter preventing damage [34]. In fact, sex hormones, including estrogen and testosterone, have been shown to regulate *arg-ii* expression in various tissues. However, estrogen seems to downregulate Arg-II expression [35]. At the same time, testosterone was reported to upregulate its expression [36], suggesting that higher Arg-II levels in various organs of female mice are unlikely to be attributable to sex hormones. However, it remains possible that sex hormones are responsible for the induction of Arg-II in male but not in female podocytes with aging. An alternative mechanism could be the more pronounced vascular rarefaction with aging in male as compared to female kidneys, which leads to more pronounced hypoxia, a known factor that is able to induce Arg-II strongly in podocytes [12]. Further research is needed to fully understand the mechanisms of sex-specific differences in Arg-II expression and its downstream effects in different tissues and cells.

Despite some conflicting results, epidemiology studies show a marked preponderance of glomerular diseases in males [6], the faster progression of chronic kidney diseases, and higher levels of proteinuria in men compared to women in older patients [5,37]. Sex-related differences in podocyte physiology and/or pathophysiology have also been described in species other than mice. In line with the findings in our current mouse model, studies in horses and cotton rats also show increased podocyte-related kidney dysfunction in males when compared to females [22,23]. Whether the reported differences in male and female animals are also related to differential Arg-II levels in podocytes remains to be elucidated.

In conclusion, this study demonstrates the role of Arg-II in sex-specific podocyte injury in aging. The results suggest that the elevation of Arg-II level in podocytes is, at least partially, responsible for age-related podocyte dysfunction and albuminuria in males. Thus, Arg-II upregulation in podocytes may provide a mechanism/explanation for the sex-related differences in chronic renal disease progression and outcomes during aging in humans.

### Limitations and Future Perspectives

Our current study has several limitations, and several questions remain to be answered in the future. First, two specific podocyte markers are selected to investigate podocytes via immunofluorescence staining, WT1 for cell counting and SNPT for co-localization studies with Arg-II. More markers, especially non-invasive methods investigating podocyte injury such as urinary podocin concentration, could be implemented and are of great diagnostic importance. Second, as discussed above, the mechanisms of the sex-specific regulation of Arg-II in different cell types of the kidneys, i.e., lower Arg-II levels in proximal tubular cells but higher expression levels in podocytes in aged males than females, remain to be investigated. It also remains to be investigated whether Arg-II could be upregulated in more advanced aged female mouse podocytes and whether *arg-ii* ablation could reduce albuminuria in these mice. Third, since Arg-II is not only upregulated in podocytes but also in other glomerulus cells in aging, the identification of these cell types that express Arg-II in aging and the analysis of these cell functions in relation to albuminuria is important. For this purpose, cell-specific animal models, including podocyte-, mesangial cell- and endothelial cell-specific *arg-ii^−/−^* mice, shall be generated and used to decipher the cell-specific contribution of Arg-II in renal aging. Fourth, it would be interesting to investigate the role of podocyte-specific Arg-II in various renal disease models or podocytopathy models. Finally, it is important to translate the findings in animal models to human patients, since the relationship between podocyte injuries and renal functions exhibit species difference [23].

## 4. Materials and Methods

### 4.1. arg-ii^−/−^ Mouse and Sample Preparation

Wild type (*wt*) and *arg-ii* knockout (*arg-ii*^−/−^) mice were kindly provided by Dr. William O’Brien (Shi et al., 2001 [38]) and backcrossed to C57BL/6 J for more than 10 generations. Genotypes of mice were confirmed via polymerase chain reaction (PCR) as previously described. The offspring of *wt* and *arg-ii*^−/−^ mice were generated via interbreeding from hetero/hetero cross. Mice were housed at 23 °C with a 12 h light–dark cycle. Animals were fed a normal chow diet and had free access to water and food. The animals were selected and included in the study according to age, sex, and genotype. Only healthy animals, based on the scoring of activity, posture, general appearance (coat, skin), locomotion, eyes/nose, and body weight loss, were used. In this study, confounders such as the animal/cage location were controlled. All experimenters were aware of the group allocation at the different stages of the experiments. Male and female mice at the age of 3–5 months (young) or 20–22 months (old) were euthanized under deep anesthesia (i.p. injection of a mixture of ketamine/xylazine at 50 mg/kg and 5 mg/kg, respectively), and death was confirmed by absence of all the reflexes and by exsanguination. Upon animal euthanization, kidneys were fixed with 4% paraformaldehyde (pH 7.0) and then embedded in paraffin for histology and immunofluorescence staining experiments. Experimental work with animals was approved by the Ethical Committee of the Veterinary Office of Fribourg, Switzerland (2020-01-FR), and performed in compliance with the guidelines on animal experimentation at our institution and in accordance with the updated ARRIVE guidelines [39].

### 4.2. Quantification of Podocytes

The number of podocytes was quantified as previously reported [40]. The number of podocytes was investigated in the kidneys of both male and female (young and old) *wt* and *arg-ii*^−/−^ mice (*n* = 10 per each experimental group). For each sex, a number of old *wt* and *arg-ii*^−/−^ mouse podocytes were compared, and respective young groups were used as the control. Podocyte nuclei were identified based on positive Wilms tumor protein 1 (WT1) expression. Immunofluorescence staining of WT1 was performed in combination with synaptopodin (SNPT), which was used to define the glomeruli border and calculate the podocyte area. Three fields of view per kidney were imaged, and podocyte nuclei were counted in 10 to 15 of the glomeruli encountered. The data are expressed as the number of podocytes per square micrometer (podocytes/μm^2^).

### 4.3. Immunofluorescence Staining

Immunofluorescence staining was performed as previously described [41]. Briefly, the kidneys of both male and female *wt* and *arg-ii*^−/−^ mice (*n* = 10 mice per group) were isolated, fixed with 4% paraformaldehyde (pH 7.0), and eventually embedded in paraffin. After deparaffinization in xylene (2 times, 10 minutes each), the sections were treated in ethanol (twice in 100% ethanol for 3 min, and twice in 95% ethanol, once in 80% ethanol for 1 min, sequentially) followed by antigen retrieval (EDTA buffer, pH 8.0) in a pressure cooker (~95–100 °C). Primary antibodies of different species were used for co-immunofluorescence staining of Arg-II/SNPT and WT1/SNPT. Transverse sections (5 μm) were blocked with mouse Ig blocking reagent (M.O.M, Vector laboratories; Newark, CA, USA) for 2 h and then with PBS containing 1% BSA and 10% goat serum for 1 h. The sections were then incubated overnight at 4 °C in a dark/humidified chamber with primary antibodies and subsequently incubated for 2 h with the following secondary antibodies: Alexa Fluor 488-conjugated goat anti-rabbit IgG (H+L) and Alexa Fluor 568-conjugated goat anti-mouse IgG (H+L). All of the sections were finally counterstained with 300 nmol/L DAPI for 5 min. Immunofluorescence signals were visualized under a Leica TCS SP5 confocal laser microscope (Leica AG; Wetzlar, Germany). The antibodies are shown in Table 1.

### 4.4. Urine Collection, Measurement of Urine Albumin and Creatinine

The urine collection was performed to evaluate the concentration of albumin and creatinine. The albumin/creatinine ratio was used as a readout of kidney damage. The urine samples of young and old *wt* and *arg-ii*^−/−^ mice of both sexes were collected on a hydrophobic sand (LabSand^®^, Coastline Global, Palo Alto, CA, USA) according to the manufacturer’s instructions [42]. Animals were deprived food, with access to water for the duration of the urine collection (12 h). Each mouse was placed alone in solid-bottom cages containing the LabSand. Urine drops were collected each hour with a pipette and transferred to a polypropylene tube with a cap.

Urine creatinine was measured via colorimetric assays using a creatinine urinary detection kit (EIACUN, Invitrogen/Thermo Fisher Scientific, Waltham, MA, USA). Briefly, urine was diluted 1 to 25 with distilled water and then incubated with creatinine reagent for the indicated time (30 min). Absorbance was read at 490 nm. Albuminuria was measured via ELISA using a mouse Albuwell kit (1011, Exocell Inc., Philadelphia, PA, USA), according to the manufacturer’s instructions [43]. Briefly, urine samples were diluted 1 to 25 with NHEBSA and incubated with anti-mouse Albumin Ab-HRP conjugate for 30 min. Subsequently, Color Developer was added to each well and incubated for 10 min. The reaction was stopped by adding Color Stopper. Absorbance was read at 450 nm. Albumin was normalized by creatinine, and albuminuria was quantified by means of the albumin/creatinine ratio (uACR, μg/mg).

### 4.5. Blood Collection and Measurement of Serum Creatinine

The blood collection was performed to evaluate serum creatinine concentration as complementary data to assess age-related glomerular dysfunction. The blood was collected from the jugular vein of young and old *wt* and *arg-ii*^−/−^ mice of both sexes (*n* = 5 per each experimental group) as previously described [44]. Briefly, animals were anesthetized with 5% isoflurane in oxygen and maintained at 1.5% isoflurane during the procedure. Blood was taken from the jugular vein with a 30 G insulin syringe (B. Braun; Melsungen, Germany). After collection, blood was transferred to a tube with a gel clot activator (Microvette 500 Z-Gel; SARSTED AG; Nümbrecht, Germany) for clotting for 15 min. After 15 min, blood was centrifuged for 5 min at 10,000× *g* to separate the serum. The serum was stored at −80 °C until use. Serum creatinine was measured by colorimetric assay using a mouse Creatinine detection kit (#80350, Crystal Chem, Elk Grove Village, IL, USA). All measurements were performed in duplicate.

### 4.6. Quantification of Glomerulosclerosis

Periodic acid-Schiff (PAS) staining was performed to evaluate age-related glomerulosclerosis in *wt* and *arg-ii*^−/−^ mice. The analysis was conducted on male and female mice (*n* = 9 per each group), and young animals were used as the control [45]. Kidney tissue samples were evaluated using light microscopy upon periodic acid–Schiff (PAS) staining (ab150680, Abcam; Cambridge, UK). PAS-positive tissue was stained magenta, and nuclei were counterstained with hematoxylin. A score of severity from 0 (no sclerosis) to 4 (maximal sclerotic atrophy) was assigned to each glomerulus. Briefly, a score of 0 reflected an unaffected glomerulus, a score of 1 indicated sclerosis involving < 25% of the glomerular tuft; a score of 2 indicated a sclerosis of 25–50%; a score of 3 represented a sclerosis of 50–75%; and a score of 4 indicated global sclerosis > 75% or the complete collapse of the glomerular tuft. Approximately *n* = 100 glomeruli per kidney were analyzed to determine the glomerulosclerosis score. For each score (0 to 4), the percentage of glomeruli with sclerotic atrophy compared with the total glomerular number was reported.

### 4.7. Statistical Analysis

Statistical analysis was performed using GraphPad Prism 9.5.0 (La Jolla, CA, USA). Data were presented as mean ± S.D. Data distribution was determined by the Kolmogorov–Smirnov test, and statistical analysis for normally distributed values was performed with analysis of variance (ANOVA) with Fisher’s LDS test. For non-normally distributed values, the Kruskal–Wallis test was used. Differences in mean values were considered significant at a two-tailed, and the following was used to define significant differences: * *p* ≤ 0.05, ** *p* ≤ 0.01, *** *p* ≤ 0.001, **** *p* ≤ 0.0001.

## Figures and Tables

**Figure 1 ijms-24-11228-f001:**
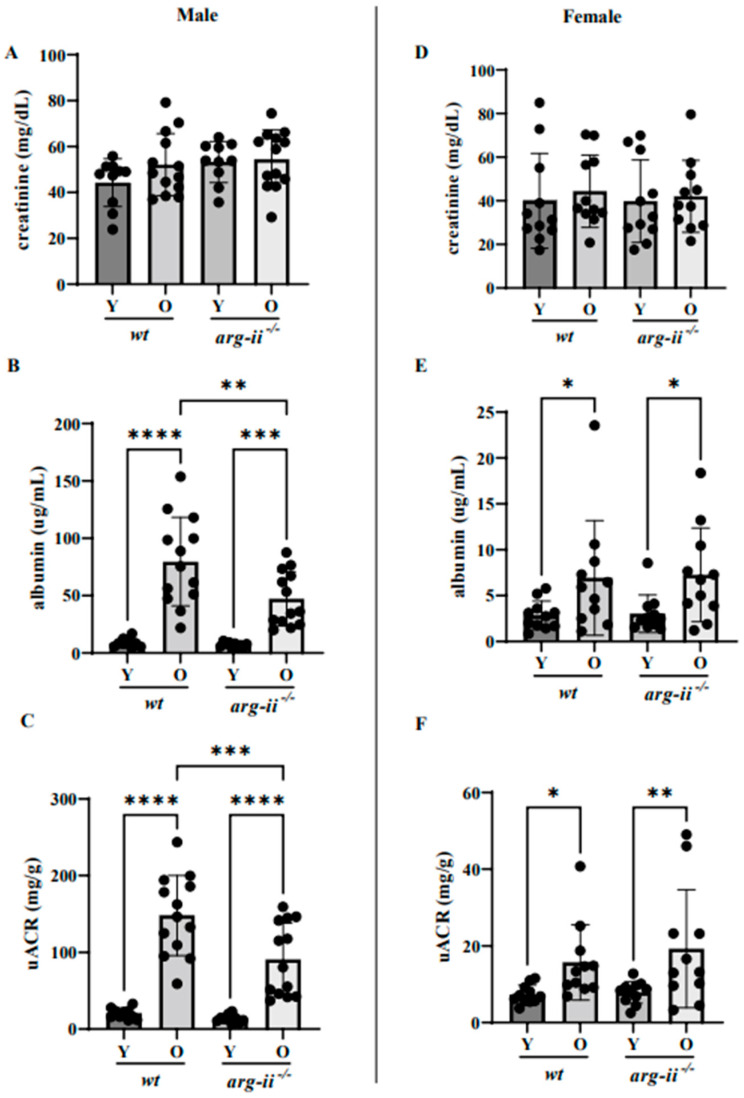
Arg-II knockout decreases albuminuria in aged male mice. Urinary creatinine (**A**,**D**) and albumin (**B**,**E**) were measured in young and old *wt* and *arg-ii^−/−^* mice of both sexes. Glomerular function (**C**,**F**) monitored via albuminuria is reported as the ratio of urinary albumin/creatinine (uACR). *n* = 11–13. * *p* ≤ 0.05, ** *p* ≤ 0.01, *** *p* ≤ 0.001, **** *p* ≤ 0.0001.

**Figure 2 ijms-24-11228-f002:**
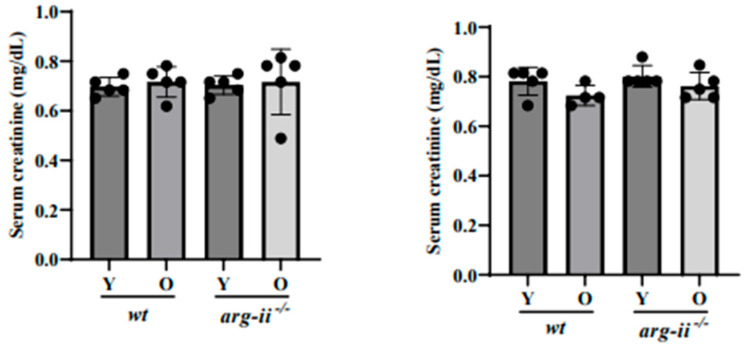
Serum creatinine is not affected by aging and sex. Serum creatinine levels were measured to evaluate glomerular dysfunction. *n* = 5.

**Figure 3 ijms-24-11228-f003:**
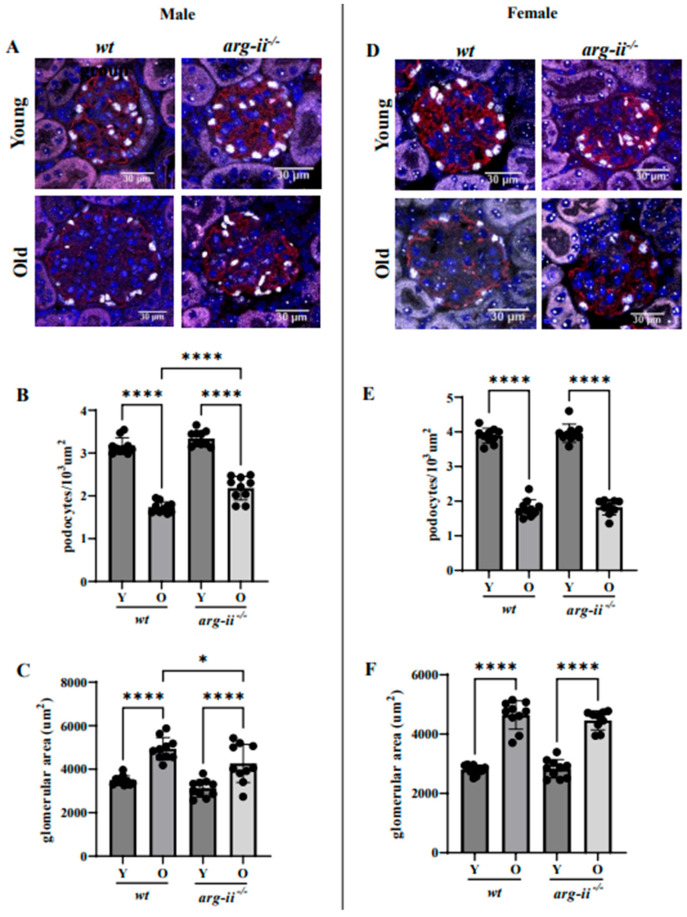
Podocyte number and density. (**A**,**D**) Representative confocal images of kidney glomeruli of all the groups showing WT1 (white) and synaptopodin (SNPT, red) co-staining. DAPI was used to stain the nuclei (blue). (**B**,**E**) Quantification of the podocyte density in glomeruli of male (**B**) and female mice (**E**). (**C**,**F**) Podocyte density was calculated as the average of all analyzed glomeruli per kidney. Scale bar: 30 µm. *n* = 10. * *p* ≤ 0.05, **** *p* ≤ 0.0001.

**Figure 4 ijms-24-11228-f004:**
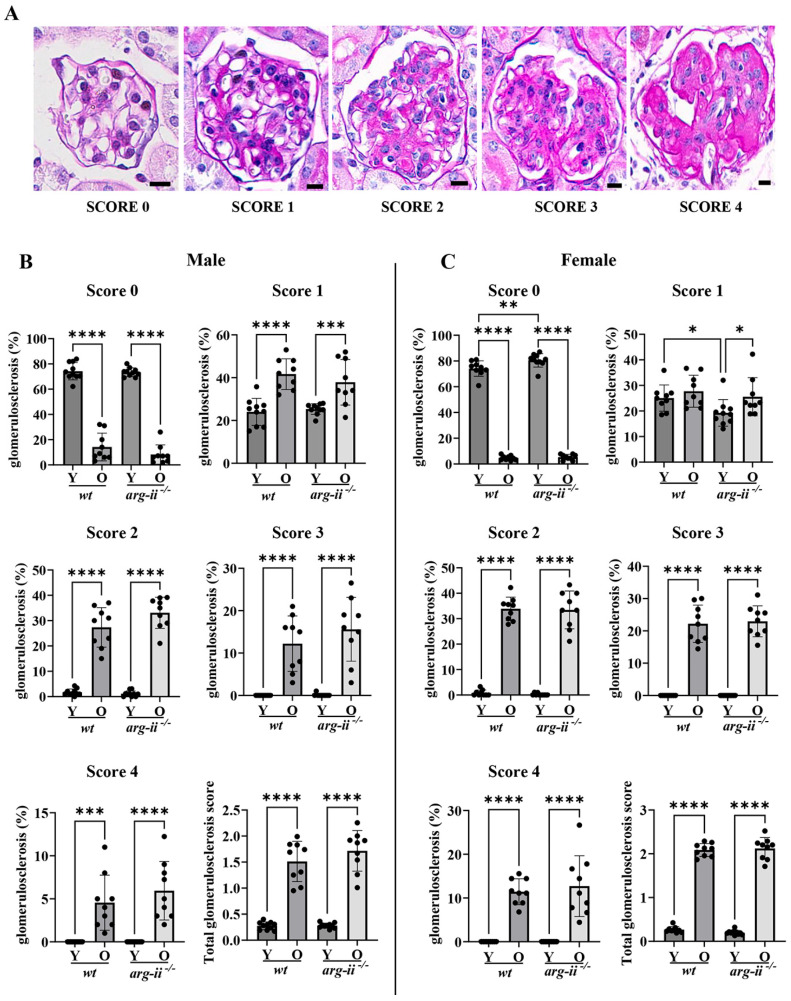
Glomerulosclerosis score. (**A**) Representative image for each score category obtained with PAS staining. PAS-positive tissue is stained magenta; nuclei are counterstained with hematoxylin (dark blue). Scale bar: 20 µm. (**B**,**C**) Quantification of glomerulosclerosis scores for male (**B**) and female mice (**C**). The total glomerulosclerosis score was calculated from the average score of 100 glomeruli per kidney. *n* = 9. * *p* ≤ 0.05, ** *p* ≤ 0.01, *** *p* ≤ 0.001, **** *p* ≤ 0.0001.

**Figure 5 ijms-24-11228-f005:**
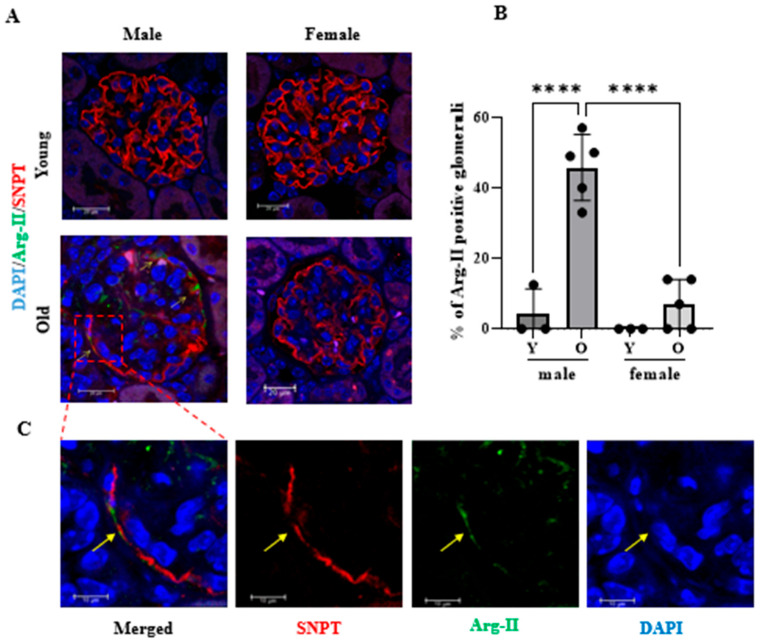
*WT* old male glomeruli show Arg-II expression, while female glomeruli do not. (**A**) Co-immunostaining of glomeruli for Arg-II (green) and SNPT (red). Nuclei were stained with DAPI (blue). A merged image is shown. Scale bar: 20 µm. (**B**) Quantification of Arg-II-positive glomeruli. (**C**) The enlarged image of the indicated insert in (**A**) reveals the co-localization of Arg-II with the podocyte marker SNPT in the glomeruli of aged male mice. Arrows indicate co-localization. Scale bar: 10 µm. **** *p* ≤ 0.0001.

**Table 1 ijms-24-11228-t001:** Antibodies and dilutions used for confocal microscopy.

Antibody Target	Dilution
Arg-II (#55003, Cell Signaling; Danvers, MA, USA)	IF 1:100
Synaptopodin (sc-515842, Santa Cruz Biotechnol; Dallas, TX, USA)	IF 1:50
WT1 (ab89901, Abcam; Cambridge, UK)	IF 1:100
Alexa Fluor 488-conjugated goat anti-rabbit IgG (H+L) secondary Ab (A-11008, Thermo Fisher Scientific; Waltham, MA, USA)	IF 1:400
Alexa Fluor 568-conjugated goat anti-mouse IgG (H+L) secondary Ab (A-11031, Thermo Fisher Scientific; Waltham, MA, USA)	IF 1:400

## Data Availability

Data supporting the present study are available from the corresponding author upon reasonable request.

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
