# Peer review of "Elevation of Arginase-II in Podocytes Contributes to Age-Associated Albuminuria in Male Mice"

_ijms, 2023, doi:10.3390/ijms241311228_

Round 1
Reviewer 1 Report
1. Everything that increases or affects kidney blood flow could cause kidney failure. There is a lot of chronic or acute injury (e.g., dehydration, nephrotoxic drugs, toxic damage to the renal tubules, heart disease).
2.There is a general opinion that in animals, the most common cause of kidney injury is toxins and/or an infectious disease. It would also be worth emphasising that the number of diseases that could cause podocyte injury increases with age.
3.The authors in the manuscript concentrate only on the mouse course of disease.
3a. Low references are included for human studies. There is absolutely no comparison of the studies of podocyte injury in animals other than mice (such as a dog or a horse).
3b. It is interesting that in male horses, podocyte injury was also present. In a group of horses, a trend of increased concentrations of podocin in males compared to females was also observed.
These sections should be improved.
4. The section limitations should include the information that only one marker for podocytes was detected. Increasing the number of markers would increase the sensitivity of the method.
Reviewer 2 Report
Comments for manuscript ijms-2369963
General comments:
The paper is methodologically well structured. The manuscript's Abstract, Introduction, M&M, and R&D consist of adequate writing of the original research article (however, minor revision of style of writing is needed, especially in R&D). Conclusion is missing in this manuscript. Pay attention to the structure of sentences and grammar. The topic is interesting and worth publishing but with revised segments. Also, the literature review should be revised in order to provide and prove the novelty of the paper. In order to publish, the manuscript needs major revisions.
Specific comments:
1. After reading the entire paper, the title should be more precisely defined
2. Also, the title should include a model of work
3. Abstract should be structured accordingly
4. abstract should be rewritten with adequate methodology
5. More emphasis needs to be placed on the introduction. Last paragraph of Introdoction section should contain research gap and highlight the purpose of the study, such as novelty.
6. Authors are advised to avoid lumping of references, and to rewrite paragraphs and -sentences with it (for example lines 47-49).
7. Authors are advised to exclude personal pronouns in Manuscript (such as we, our etc.) and transform Manuscript into Passive voice.
8. In the part … by the Ethical Committee of the Veterinary 204 Office of Fribourg Switzerland (2020-01-FR) and performed in compliance with guidelines 205 on animal experimentation at our institution … authors have to indicate reference for used guidelines at their institute, and refer if the used Guideline is according to ARRIVE Guideline. Authors have to highlight if experimental work is done according to ARRIVE G.
9. Animal model description lacks the essential data for in vivo testing according to GLP, have to be improved.
10. References for all standard procedures should be added.
11. The following sentence from the text: Please pay attention to scales … is not according to medical and manuscript writing form – dismiss such forms of writing.
12. Difficult to follow with the correlation to previously presented results.
13. Too much general data presented
14. Authors are advised to check the quality of figures, as most of them are blur. Figure 5, should be better explained.
15. No limitations and future perspectives of the study presented
16. No obvious or relatable comparison with previously published papers
17. Not critically justified
18. Separate conclusion is missing in the paper
19. Address conclusion accordingly
20. Major revision of references is needed – refresh with novel and up-to-date references
21. More than 50% of references used in the paper are older than 3-4 years
22. At least 15% references are older than 10 years
Has to be improved
